# Diversity of Fecal Indicator Enterococci among Different Hosts: Importance to Water Contamination Source Tracking

**DOI:** 10.3390/microorganisms11122981

**Published:** 2023-12-14

**Authors:** Soichiro Tamai, Yoshihiro Suzuki

**Affiliations:** Department of Civil and Environmental Engineering, Faculty of Engineering, University of Miyazaki, Miyazaki 889-2192, Japan; hh17057@student.miyazaki-u.ac.jp

**Keywords:** *Enterococcus*, bacterial diversity, host-specific bacteria, fecal source tracking, water quality

## Abstract

*Enterococcus* spp. are common bacteria present in the intestinal tracts of animals and are used as fecal indicators in aquatic environments. On the other hand, enterococci are also known as opportunistic pathogens. Elucidating their composition in the intestinal tracts of domestic animals can assist in estimating the sources of fecal contamination in aquatic environments. However, information on the species and composition of enterococci in animal hosts (except humans) is still lacking. In this study, enterococci were isolated from the feces of cattle, pigs, birds, and humans using selective media. Enterococcal species were identified using mass spectrometry technology, and each host was characterized by diversity and cluster analysis. The most dominant species were *E. hirae* in cattle, *E. faecium* in birds, and *E. faecalis* in pigs and humans. Cattle had the highest alpha diversity, with high interindividual and livestock farm diversity. The dominant enterococcal species in pigs and humans were identical, and cluster analysis showed that the majority of the two hosts’ species clustered together.

## 1. Introduction

Enterococci are Gram-positive lactic acid bacteria commonly found in the intestinal tracts of all animals, from insects to humans. They are resistant to a variety of environmental factors, such as alkalis, pH fluctuations, temperature increase, and high sodium chloride concentration, and have a longer half-life in the environment compared to other intestinal bacteria [1,2]. Because enterococci are present in large amounts in the feces of warm-blooded animals and do not multiply easily in nature, they are used as fecal indicators in aquatic environments, such as rivers, lakes, and marshes [3,4]. Due to contamination inputs, these bacteria are commonly detected in rivers worldwide and are, therefore, ubiquitous in the human living environment [5,6,7,8,9,10].

Although enterococci are not highly pathogenic, they are known to cause opportunistic infections in immunocompromised individuals, such as infants, hospitalized patients, and the elderly [11]. In recent years, these bacteria have become an increasingly common cause of serious nosocomial infections in the United States and Europe [12]. This is because they are naturally resistant to various antibiotics and, with the increase in the number of antibacterial drugs, multidrug-resistant enterococci have emerged, making their treatment more difficult [13,14]. Among these new forms, vancomycin-resistant enterococci (VRE), which are resistant to vancomycin, an antibiotic that was originally considered particularly effective against enterococcal infections, have been cited as one of the most important groups of antibiotic-resistant intestinal bacteria [15,16,17]. Although their isolation has been less reported in clinical settings in Japan than in Western countries, the diffusion of nosocomial infections caused by VRE is a concern [18,19]. Enterococci can cause infections not only in humans but also in companion animals, such as dogs and cats, and livestock animals, such as cattle, pigs, and birds [20].

On the other hand, the global demand for meat is on the rise, and the livestock industry is expected to continue to develop [21]. However, such development raises concerns related to groundwater and surface waters being potentially contaminated by livestock wastewater [22]. The main species of enterococci differ depending on the host animals. For example, *E. faecalis* and *E. faecium* are the most abundant species in the human intestinal tract, along with other species, such as *E. durans*, *E. avium*, and *E. gallinarum* [23]. Therefore, identifying the enterococcal species present in each host and elucidating their characteristics may assist in research efforts aiming, for example, at the identification of the sources of fecal contamination and infectious diseases in aquatic environments.

Enterococci have conventionally been identified by sequencing the 16S rRNA gene, but this method is complex, labor-intensive, and costly in terms of specimen processing and testing [6,24,25]. In addition, it is sometimes difficult to accurately identify species by sequencing only the V3–V4 region [25,26]. For these reasons, only a few surveys of host and enterococcal species have been conducted, and even today there is little information on these bacterial species, especially in animals (except for humans). Studies comparing enterococcal species carried by different individuals belonging to the same animal host are even more scarce. Therefore, it is important to accumulate information on enterococci from different host sources. Matrix-assisted laser desorption/ionization time-of-flight mass spectrometry (MALDI-TOF MS) is a method used to identify bacterial proteins by ionizing them and matching their mass spectral patterns with a database. Databases collecting information on bacteria have also been created, and MALDI-TOF MS has become a mainstream method for the identification of bacterial species. It is a useful and cost-effective technology because of its simplicity and rapid operation protocol, from sample preparation to identification [27,28]. In this study, we investigated the presence of enterococci in fecal samples obtained from cattle, pigs, birds, and humans using MALDI-TOF MS and characterized each host by diversity and cluster analysis to understand the enterococcal composition in each host.

## 2. Materials and Methods

### 2.1. Collection of Fecal Material from Each Host

The fecal samples used in this study were collected from livestock farms in the Southern Kyushu region, Japan, which has the most active livestock industry in the country and is a production center for cattle, pigs, and birds (broilers). Fecal samples from domestic animals were collected directly from the excrement of individual animals in sterile tubes immediately after excretion. Table 1 shows the number of isolates collected from each host: cattle feces were collected from 8 individuals at Farm A, 1 individual at Farm B, and 1 individual at Farm C; pig feces were collected from 11 individuals at farm D; and bird feces were collected from 10 individuals (6 weeks old) at poultry farm E. For human feces, 10 samples were collected in sterile tubes by wiping the feces from the edge of a Japanese-style toilet bowl (the part not in contact with water) with a sterile cotton swab. The probability that the fecal samples collected in this study were contaminated with feces from other individuals or animals is extremely low. Human feces were collected from male public toilets used by an unspecified number of people in a public facility.

### 2.2. Isolation of Enterococci

The fecal material collected from each host was suspended in PBS buffer (NIPPON GENE, Tokyo, Japan) and filtered through a mixed cellulose ester membrane (47 mm in diameter, 0.45 µm in pore size, Advantec, Tokyo, Japan). Filters were applied to membrane-Enterococcus Indoxyl-β-d-Glucoside agar (mEI medium, Becton, Dickinson and Company, Bergen, NJ, USA), a selective medium for enterococci, and incubated at 41 °C for 24 h [29]. Blue colonies that grew on the filter after incubation were isolated as enterococcal isolates. The isolates were placed in Todd–Hewitt agar medium (agar 1.5%, Becton, Dickinson and Company) to obtain pure colonies, incubated at 37 °C ± 0.5 °C for 24 h, and stored. Overall, 50 enterococcal isolates were obtained from three individuals belonging to the sampled cattle, pigs, and birds, and 10 were obtained from each of the remaining individuals. Human-derived enterococci were less abundant than those observed in the other hosts; thus, all growing colonies were isolated (Table 1).

### 2.3. Identification of Bacterial Species via MALDI-TOF MS

The obtained isolates were analyzed using MALDI-TOF MS (microflexLT/SH, Bruker Daltronics, Billerica, MA, USA) [30]. In brief, the isolates were cultured in Brain Heart Infusion (Becton, Dickinson and Company) liquid medium at 37 °C for 24 h. After incubation, each isolate was centrifuged at 13,000 rcf for 5 min, and the supernatant was removed. A total of 300 µL of Milli-Q water and 700 µL of ethanol (Wako Pure Chemical Industries, Osaka, Japan) were added, and the solution was vortexed. Each sample was left standing for at least 10 min and was then centrifuged at 13,000 rcf for 5 min at room temperature. After removing the supernatant, 50 µL of 70% formic acid and acetonitrile (Wako Pure Chemical Industries) were added to each pellet and mixed. After centrifugation at 8000 rcf for 3 min, 1 µL of the supernatant was spread thinly on a target plate, which was left to air-dry for 10 min. Then, 1.0 µL of α-Cyano-4-hydroxycinnamic acid matrix solution (Bruker Daltonics, Billerica, MA, USA) was overlaid on the target plate. The Bruker FlexContorol 3.0 was used for the measurements with a linear positive 3–20 kDa. The acquisition range of the mass-to-charge ratio, number of integrations, detector gain, and laser power were set to 2000–20,000 Da, 5000–10,000, 1900 V, and 65%, respectively. A Bruker bacterial test standard (part no. 8255343, Bruker Daltronics) was used for instrument calibration [30]. After calibration, the mass spectra obtained by ionizing molecules contained in bacteria by laser irradiation were checked against a database of spectral information on a large number of known bacteria by Biotyper 3.1.6 (Bluker Daltonics) [31,32]. The spectral data of unknown microorganisms obtained from the measurement were calculated as a peak list and compared to the database’s peak list. The comparison with the database generates a log(score) on a 3-point scale, where a higher score means a higher similarity between the unknown microorganisms and the reference library’s peak list. Enterococcal species with a high match rate with a score of 1.8 or higher were selected.

### 2.4. Statistical Analysis

Statistical analyses were performed in R (4.3.0). The alpha diversity of each sample was determined using the diversity function of the vegan R package and the Shannon index. Beta diversity and clustering among samples were determined based on the Bray–Curtis distance, and the former was visualized using a principal coordinate analysis plot.

## 3. Results and Discussion

### 3.1. Analysis Based on the Bacterial Composition of Each Host

Figure 1 shows the results of the alpha diversity analysis of each host based on the Shannon index. Birds’ feces, where *E. faecium* was dominant overall, had the lowest alpha diversity, while cattle’s feces, where the dominant species differed from individual to individual, had the highest alpha diversity. 

Beta diversity analysis based on the Bray–Curtis distance was performed to assess the similarity of the enterococcal bacterial composition among hosts (Figure 2). While the cattle and bird samples were clustered together in single groups, the pig samples were grouped in two distinct clusters separated from each other, reflecting the polarization of dominant species within this host. The human samples also clustered in a single group.

In the plot, an overlap was detected between pig and human samples, which indicated a high similarity in the composition of enterococci in these two hosts.

Therefore, cluster analysis was used to further evaluate the similarity among bacterial compositions (Figure 3). Cluster-A consists of only one human sample. Cluster-B is composed of isolates from all the hosts, showing that some enterococcal species were shared. Cluster-C is a distinct cattle cluster with six out of seven individuals being cattle. Similarly, cluster-D consists mainly of birds. Cluster-E is constituted by isolates from the human and pig hosts, highlighting the similarity in the enterococcal compositions between the two hosts. It has already been reported that the intestinal bacteria of humans and pigs are similar [33,34], and our results also support this finding with respect to enterococci. Overall, based on the similarity of their enterococcal composition, the four hosts examined in this study can be broadly classified into three categories: humans and pigs, cattle, and birds.

### 3.2. Identification of Enterococcal Species in Each Host

Figure 4a shows the relative abundance of enterococcal species detected in cattle feces. A total of 220 isolates isolated from the feces of 10 individuals were identified by MALDI-TOF MS, with *E. hirae* (154 isolates) being the most abundant, accounting for 70%, followed by *E. casseliflavus* (30 isolates, 13.6%) and *E. saccharolyticus* (16 isolates, 7.27%). *E. mundtii* (seven isolates) and *E. gallinarum* (four isolates) were also detected. *E. saccharolyticus* and *E. mundtii* were specific to cattle among the hosts examined in this study. However, the detection rates of *E. faecium* (eight isolates) and *E. faecalis* (one isolate), which are known as major enterococcal species inhabiting cattle intestinal tracts, were low. A total of seven enterococcal species were identified from each cattle individual, the highest number detected among all hosts. Cattle in Farm A shared the following dominant species: *E. hirae* (three), *E. saccharolyticus* (two), *E. casseliflavus* (one), *E. faecium* (one), and *E. hirae* and *E. saccharolyticus* (one). *E. casseliflavus* was dominant in both Farm B and Farm C. Even within the same livestock farms, the dominant species differed among individuals. These results indicated that cattle harbored a wide variety of enterococci and that the dominant species greatly varied depending on the sampling location. This may be attributed to differences in diet, geographical factors, and farming practices. Therefore, it is necessary to know the enterococcal species of livestock on nearby farms when estimating the source of fecal contamination in the aquatic environment. *E. casseliflavus* was detected in the largest number of individuals (seven), regardless of the livestock farms or time of collection. This species was detected in other hosts, but the highest detection rate was in cattle, suggesting that it is a major species for cattle. Although *E. durans* has also been reported as one of the common enterococcal species found in cattle [35], it was not detected in this study. This may be due to the fact the 16S rDNA gene of *E. durans* shows a high similarity of more than 98.8% with *E. hirae*, making it difficult to distinguish between the two species via genetic analysis [36]. As *E. hirae* was detected in large numbers in this study, it is possible that some of the *E. durans* strains identified from cattle via genetic analysis in previous studies may be actually identified as *E. hirae* when analyzed using MALDI-TOF MS. *E. hirae* has an extremely high detection rate in cattle but low detection rates in humans or in aquatic environments. [37]. In this study, they were detected in all hosts, but the results may be difficult to use in tracing sources of contamination in aquatic environments, as their viability in the aquatic environment is expected to be low. Unlike *E. faecalis* and *E. faecium*, *E. hirae* has a low detection rate of virulence genes and is rarely a causative agent of infectious diseases in humans [37]. Enterococci do not multiply easily in nature; the dominance of *E. hirae* in public waters near farms is most likely due to contamination by cattle or other animals. Identifying the source of contamination by environmental DNA analysis using genetic markers of animals is difficult because these genes can be detected even in properly treated water. In tracking the source of contamination, it is important to identify the major bacterial species of the host and conduct a culture-based analysis.

Figure 4b shows the relative abundance of enterococcal species in pig feces. A total of 230 isolates were identified from 11 pigs using MALDI-TOF MS, with *E. faecalis* (140 isolates) exhibiting the highest abundance (60.9%), followed by *E. hirae* (73 isolates, 31.7%). *E. villorum* was detected only in pigs among the four hosts examined. While it has indeed been reported only in pigs [38], this species was detected in only one of the eleven individuals examined, indicating that its prevalence was low. The comparison of bacterial composition among individuals showed that the most dominant species were *E. faecalis* in six pigs, *E. hirae* in four pigs, and both *E. faecium* and *E. villorum* in one pig. In previous studies, *E. faecium* was detected at extremely high frequencies, while *E. faecalis* was detected most frequently in the present study [39,40]. However, results were consistent with *E. faecalis*, *E. faecium*, and *E. hirae* being identified as the top three species. The dominant species differed among the pigs examined in this study, even though they were reared on the same farm, but bacterial compositions were expected to be dominated by the three aforementioned enterococcal species. 

Figure 4c shows the relative abundance of enterococcal species in bird feces. A total of 219 isolates were identified from 10 birds using MALDI-TOF MS, with *E. faecium* (173 strains) exhibiting the highest abundance (79.0%), followed by *E. faecalis* (35 isolates, 16.0%), *E. hirae* (10 isolates), and *E. casseliflavus* (1 isolate). These four bacterial species were commonly detected in all the hosts examined in this study. The comparison of bacterial compositions among individuals revealed that *E. faecium* was the most dominant species in the majority of individuals (eight birds), while *E. faecalis* and *E. hirae* were both dominant in only one individual. *E. faecalis* and *E. faecium* have been previously reported as the most common enterococci isolated from broilers [41], which is in line with the results of the present study. However, here, the isolation frequency of *E. faecalis* was much lower than that of *E. faecium*. Studies have shown that broilers are dominated by *E. faecalis* at a young age and, as they grow older, *E. faecium* becomes the dominant species [42,43]. Since all broilers in this study were 6 weeks old, which is an advanced age for broilers, *E. faecium* was expected to be the most dominant species. In a previous study determining enterococcal species in broiler feces from nine farms by differences in enzyme activity, *E. faecium* was the most dominant, and there were no significant differences in the distribution of enterococcal species among the farms [44]. It was suggested that the dominant enterococci in broilers raised on the same farm were similar.

Figure 4d shows the relative abundance of enterococci species in human feces. A total of 142 isolates were identified from 10 samples using MALDI-TOF MS, with *E. faecalis* (71 isolates) exhibiting the highest abundance (50.0%), followed by *E. casseliflavus* (26 isolates, 18.3%) and *E. hirae* (19 isolates, 13.4%). *E. durans* and *E. avium* were detected only in humans among the four hosts examined in this study. Seven bacterial species identified in human feces were equally common in cattle feces. The comparison of bacterial compositions among samples revealed that *E. faecalis* was the most dominant species in 6 out of 10 samples, accounting for more than 90% of the total enterococci in 5 of them. The most dominant species in the remaining four samples were *E. hirae* (86.4%), *E. casseliflavus* (70%), *E. gallinarum* (66.7%), and *E. durans* (100%). Humans have an extremely diverse diet compared to other domestic animals, suggesting that the gut microbiome differs greatly among hosts [45,46].

In previous studies, *E. faecalis* and *E. faecium* have been reported as the most abundant enterococci isolated from human feces, with a trend of interindividual variation in other enterococci species [18,19,47]. However, in this study, while *E. faecalis* was detected in a large number of samples, *E. faecium* was present in only three samples and at a low frequency (4.5–9.1%). This trend was not observed in previous studies and was particular to this study. The fact that feces were collected from toilet bowls rather than rectal swabs may have weeded out enterococci with low viability. *E. hirae* and *E. durans* have been reported to occur at low rates in the human intestinal tract, and in a particular case, they have been isolated from 9% and 18% of healthy individuals, respectively [48]. In the present study, they were detected as the dominant species, which is consistent with previous findings.

The enterococcal species in human feces varied widely from weekly to monthly intervals [44]. During 180-day monitoring of enterococcal species in human feces, *E. faecalis* was the dominant species, accounting for 91% of all enterococcal strains, while *E. faecium* was not detected. However, the occupancy rate of *E. faecalis* decreased week by week, and after 30 days, it was 50%. After 180 days, *E. faecalis* was undetectable, with *E. faecium* becoming the dominant species. In the human intestine, the species of *E. faecalis* and *E. faecium* are the main enterococcal species and continue to change along with other species due to daily dietary habits. In rivers into which treated wastewater or domestic wastewater flows, the proportion of *E. faecalis* and *E. faecium* is approximately 30%, and the most dominant species tend to be replaced by *E. faecalis* and *E. faecium* [49]. The results of fixed-point monitoring in one river showed that the occupancy rate of *E. faecium* was higher than that of *E. faecalis* [49]. Another report of enterococci in urban rivers indicates that the major species are *E. faecalis* and *E. faecium* and that *E. casseliflavus* is a frequently detected species [2]. In addition, *E. gallinarum* was detected, although it is a minor species. In rivers, *E. faecalis* and *E. faecium* are the major species, and other enterococcal species are thought to form the flora. Characterization of the major species by host will enable more in-depth analysis of fecal contamination surveys. The river was considered to have an anthropogenic load due to the inflow of wastewater from households without sewerage systems. In this study, *E. casseliflavus* was detected in human and cattle feces. There are no cattle farms around the watershed of this river, suggesting contamination of the river by human feces. Thus, when estimating hosts by enterococcal species in rivers, it is necessary to consider the characteristics of the riverine environment. However, since this study only included four enterococcal species, the more important species may have been missed. More accurate estimation requires species identification using TOF-MS analysis.

There are several limitations to this study. First, the number of farms and sampling sites is limited. All fecal collection points were in Japan and, excluding cattle, were collected from a single farm, which may have introduced bias. The results of this study also indicate that it is difficult to identify host animals based on bacterial species information alone. However, it is possible to understand the fate of enterococci by understanding land use, e.g., sewage treatment plants, drainage canals, and the type of livestock farms located in the area. Furthermore, by considering previous studies together, the characteristics of each host could be ascertained.

## 4. Conclusions

In this study, enterococci species were identified from feces of cattle, pigs, birds, and humans to characterize the enterococcal composition of each host. Although the enterococcal diversity found in cattle feces was high, the dominant species was *E. hirae*. *E. saccharolyticus* and *E. mundtii* were detected only in cattle, suggesting that these are cattle-specific enterococci, being potentially useful in identifying sources of pollution in river water. Overall, *E. faecalis* was the dominant species in the pig hosts, despite the interindividual diversity found even though the samples were collected from the same livestock farms. The dominant species in birds was *E. faecium* and showed the lowest alpha diversity. *E. faecalis* and *E. faecium* are usually found in humans, so even if a river is contaminated by them, it cannot be determined whether the source of contamination is pigs or birds. 

Furthermore, diversity and cluster analyses revealed the similarity of enterococcal species between pig and human hosts. 

The results of this study provide information on the enterococci present in a variety of animal hosts, which may assist in the identification of potential sources of fecal contamination in aquatic environments and infectious disease outbreaks. These are important One Health approaches to prevent the spread of zoonotic diseases and antibiotic-resistant bacteria.

## Figures and Tables

**Figure 1 microorganisms-11-02981-f001:**
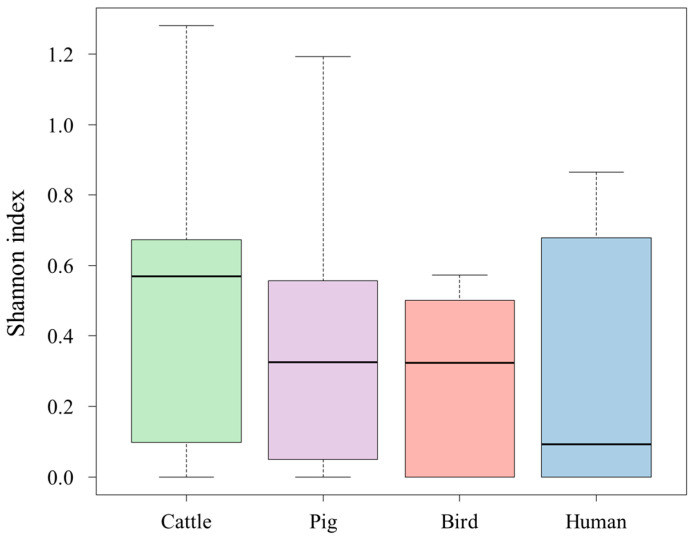
Alpha diversity analysis of each host based on the Shannon index.

**Figure 2 microorganisms-11-02981-f002:**
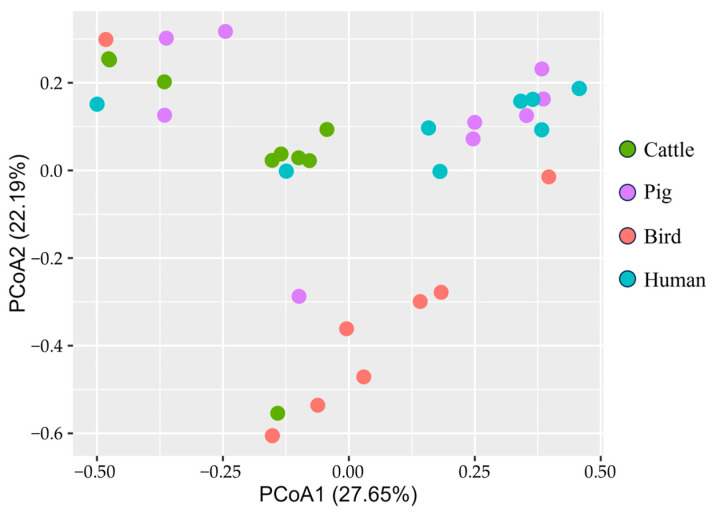
Beta diversity analysis of each host based on the Bray–Curtis distance.

**Figure 3 microorganisms-11-02981-f003:**
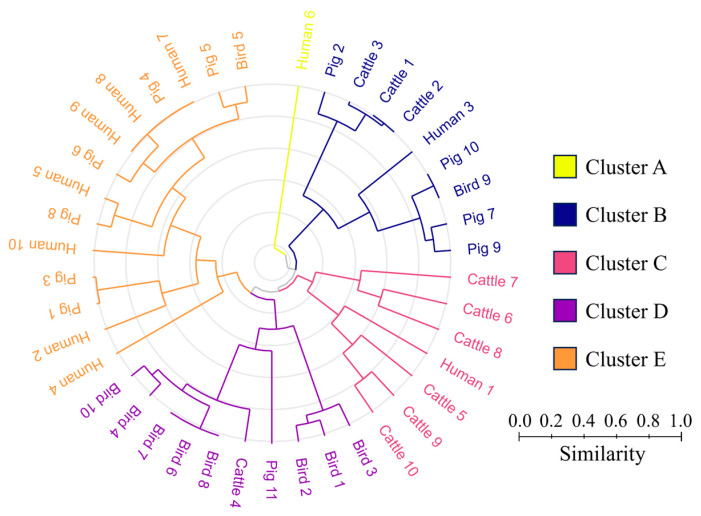
Cluster analysis based on the Bray–Curtis distance.

**Figure 4 microorganisms-11-02981-f004:**
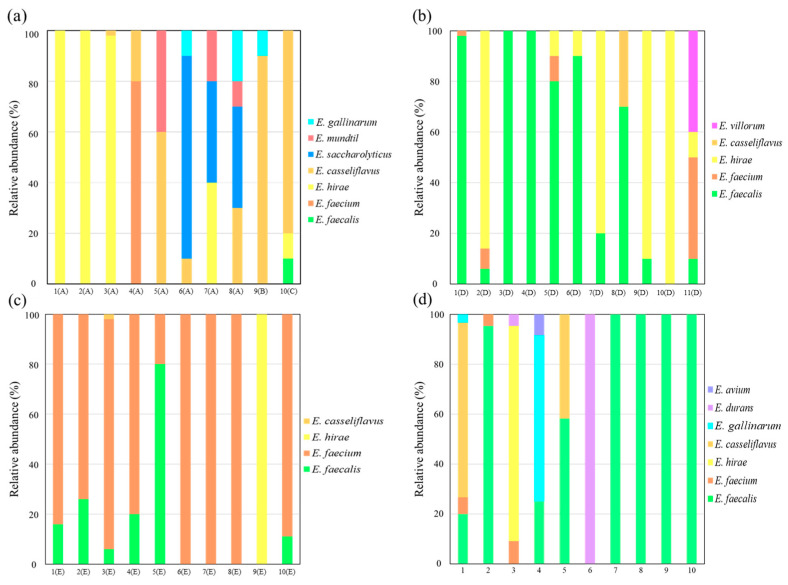
MALDI-TOF MS results showing the relative abundance of enterococcal species in the feces of each host: (**a**) cattle, (**b**) pigs, (**c**) birds, and (**d**) humans. The *x* axis indicates the individual number and farm. Farm A, A; Farm B, B; Farm C, C; Farm D, D; Farm E, E.

**Table 1 microorganisms-11-02981-t001:** Number of samples collected from each host. The number of samples analyzed is the number of individuals. Alphabets in parentheses indicate farms from which feces were sampled. Farm A, A; Farm B, B; Farm C, C; Farm D, D; Farm E, E.

	Number of Isolates Collected										
Individual Number	1	2	3	4	5	6	7	8	9	10	11
Cattle	50 (A)	50 (A)	50 (A)	10 (A)	10 (A)	10 (A)	10 (A)	10 (A)	10 (B)	10 (C)	-
Pig	50 (D)	50 (D)	50 (D)	10 (D)	10 (D)	10 (D)	10 (D)	10 (D)	10 (D)	10 (D)	10 (D)
Bird	50 (E)	50 (E)	50 (E)	10 (E)	10 (E)	10 (E)	10 (E)	10 (E)	10 (E)	9 (E)	-
Human	30	22	22	12	12	10	10	10	10	4	-

## Data Availability

Data will be made available on request.

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
