# Peer review of "Diversity of Fecal Indicator Enterococci among Different Hosts: Importance to Water Contamination Source Tracking"

_microorganisms, 2023, doi:10.3390/microorganisms11122981_

Round 1
Reviewer 1 Report
Comments and Suggestions for Authors
The manuscript describes an investigation of enterococcus spp. in four different hosts, including pig, cattle, bird and human with a goal of understanding the source of fecal contamination in aquatic environments.
First at all, I found the title is miss-leading because I am expecting to see fecal indicators study in aquatic environments. However, the research did not include any sample collection, enterococci isolation or source identification in aquatic environment. After reading the manuscript, I also found the outcomes of the research are unlikely to contribute to the effort of fecal pollution source identification in the aquatic environment because, as shown in Fig 3 and Fig 4, the bacterial species are mixed in hosts. Identification of a specific species of the Enterococcus cannot assign the species to a specific source of contamination. Authors should demonstrate, using field samples, how the outcomes of this study can be applied to fecal contamination source identification in the aquatic environment.
My second concern of this manuscript is the methodology. The manuscript cites MALDI-TOF MS as an efficient and inexpensive method for identifying bacteria to the species level. I looked up reference 19 in the manuscript, where the method was reviewed. I found there is not sufficient information to evaluate the sensitivity and specificity of the methods for identifying bacterial isolates from diverse hosts. The manuscript did not give any description of the method regarding the specificity and sensitivity nor present any information regarding the output of the results. The manuscript cited a previous paper (reference 21) led the same author of this manuscript as the evidence for its environmental application. I feel that I cannot judge the rigor of this research paper due to the lack of details of the method and primary data output from the method.
My third concern of this paper is the sample size and geographical limitation. Ten to eleven samples were taken from each host in the farms from the region for isolation of Enterococcus spp.. The analysis revealed dominant species in animal host but showed Enterococcus spp. in human hosts were more diverse. The geographic location, animal diet, farming practices may have important impact on the gut microbiome. Samples from different geographic regions (or existing isolates from other labs) and even different continents should be included before the conclusion of the dominant species can be made.
In summary, the data presented in the manuscript did not support the goal of the study. The method lacks the detail. The limitation of sample size also makes the conclusion pre-mature.
Comments on the Quality of English LanguageThe quality of English is fine.
Author Response
Thank you for reviewing our paper.
Please see the attached response sheet.

Reviewer 2 Report
Comments and Suggestions for Authors
Authors have analyzed the species characterization of Enterococcus spp. from four different animal hosts. As enterococci is an important fecal indicator bacterium counting their colonies on surface water cannot identify the source of contamination. But, for risk assessment and remediation of contamination, identification of the microbial source of contamination is important. therefore, the authors had done excellent experiments, and are able to see some trends for species patterns on some hosts. Therefore, the study has high implications for water quality monitoring and for other studies. However, the authors have not discussed the application of their study on water quality monitoring.
Further, I have noticed the following potential correction from my side.
Line 9- you mean opportunistic pathogen?
Line 31- may be no need to say developed and developing countries.
Line 32- please add, that Enterococci is an important fecal indicator for monitoring the microbial quality of water- please find other references yourself- some examples are here Doi: 10.2166/wh.2018.293, https://doi.org/10.3390/ijerph18115513, https://doi.org/10.3389/fpubh.2019.00269, Doi:10.1016/j.scitotenv.2022.160340.
Line 58-60: missing references- find other reference yourselves but potential reference can be- Doi: 10.2166/wh.2018.293, https://doi.org/10.3390/ijerph18115513, https://doi.org/10.3389/fpubh.2019.00269, Doi:10.1016/j.scitotenv.2022.160340
Comments on the Quality of English Language
moderately good
Author Response

(The authors gave the same response as above.)

Reviewer 3 Report
Comments and Suggestions for Authors
General comments
The manuscript needs some moderate language editing, especially regarding the terminologies used.
In the abstract, the authors must specify what method was used to isolate and characterise Enterococcus.
In the materials and methods, the authors must indicate who the animal samples were collected. Were these rectal swabs? Were these droppings? IF droppings, how did the authors ensure that the sample was void of environmental contaminants? How did the authors ensure that the samples were from different animals?
The conclusion must be reduced to address only the salient points.
The authors must include ethical clearance for the study.
Specific comments
The title must be revised. It is confusing. Why include the aquatic environment in the title when enterococci were only isolated from faeces of cattle, pigs, birds, and humans?
Line 14: Delete "by"
Line 74: What characteristics were determined?
Line 77 and Table 1: This should be the number of samples collected and not the number of "strains"
Line 89 onward: You should refer to "isolates" and not "strains" except if you have done strain typing.
Line 84-86: This procedure was carried out to obtain pure colonies and not to isolate the bacteria. Isolation was already done on the membranes.
Line 97: enterococcal isolates were obtained from
Line 103: Delete "The preparation steps are reported below."
Line 108: At what temperature?
Lines 109-111: Please rephrase
Line 113: Which template?
Line 147: 16S rDNA should be in italics.
Comments on the Quality of English Language
Moderate language editing is required
Author Response

(The authors gave the same response as above.)
